# Initialization of ReLUs for Dynamical Isometry

**Rebekka Burkholz**
Department of Biostatistics
Harvard T.H. Chan School of Public Health
655 Huntington Avenue, Boston, MA 02115
rburkholz@hsph.harvard.edu

**Alina Dubatovka**
Department of Computer Science
ETH Zurich
Universitätstrasse 6, 8092 Zurich
alina.dubatovka@inf.ethz.ch

## Abstract

Deep learning relies on good initialization schemes and hyperparameter choices prior to training a neural network. Random weight initializations induce random network ensembles, which give rise to the trainability, training speed, and sometimes also generalization ability of an instance. In addition, such ensembles provide theoretical insights into the space of candidate models of which one is selected during training. The results obtained so far rely on mean field approximations that assume infinite layer width and that study average squared signals. We derive the joint signal output distribution exactly, without mean field assumptions, for fully-connected networks with Gaussian weights and biases, and analyze deviations from the mean field results. For rectified linear units, we further discuss limitations of the standard initialization scheme, such as its lack of dynamical isometry, and propose a simple alternative that overcomes these by initial parameter sharing.

## 1 Introduction

Deep learning relies critically on good parameter initialization prior to training. Two approaches are commonly employed: random network initialization [4, 7, 14] and transfer learning [26] (including unsupervised pre-training), where a network that was trained for a different task or a part of it is retrained and extended by additional network layers. While the latter can speed up training considerably and also improve the generalization ability of the new model, its bias towards the original task can also hinder successful training if the learned features barely relate to the new task. Random initialization of parameters, meanwhile, requires careful tuning of the distributions from which neural network weights and biases are drawn. While heterogeneity of network parameters is needed to produce meaningful output, a too big variance can also dilute the original signal. To avoid exploding or vanishing gradients, the distributions can be adjusted to preserve signal variance from layer to layer. This enables the training of very deep networks by simple stochastic gradient descent (SGD) without the need of computationally intensive corrections as batch normalization [8] or variants thereof [12]. This approach is justified by the similar update rules of gradient back-propagation and signal forward propagation [20]. In addition to trainability, good parameter initializations also seem to support the generalization ability of the trained, overparametrized network. According to [3], the parameter values remain close to the initialized ones, which has a regularization effect.

An early example of approximate signal variance preservation is proposed in [4] for fully connected feed forward neural networks, an important building block of most common neural architectures. Inspired by those derivations, He et al. [7] found that for rectified linear units (ReLUs) and Gaussian weight initialization $w \sim \mathcal{N}(\mu, \sigma^2)$ the optimal choice is zero mean $\mu = 0$, variance $\sigma^2 = 2/N$ and zero bias $b = 0$, where $N$ refers to the number of neurons in a layer. These findings are confirmed by mean field theory, which assumes infinitely wide network layers to employ the central limit theorem and focus on normal distributions. Similar results have been obtained for $tanh$ [16, 18, 20], residual networks with different activation functions [24], and convolutional neural networks [23]. The same

derivations also lead to the insight that infinitely wide fully-connected neural networks approximately learn the kernel of a Gaussian process [11]. According to these works, not only the signal variance but also correlations between signals corresponding to different inputs need to be preserved to ensure good trainability of initialized neural networks. This way, the average eigenvalue of the signal input-output Jacobian in mean field neural networks is steered towards 1. Furthermore, a high concentration of the full spectral density of the Jacobian close to 1 seems to support higher training speeds [14, 15]. This property is called dynamical isometry and is better realized by orthogonal weight initializations [19]. So far, these insights rely on the mean field assumption of infinite layer width. [6, 5] have derived finite size corrections for the average squared signal norm and answered the question when the mean field assumption holds.

In this article, we determine the exact signal output distribution without requiring mean field approximations. For fully-connected network ensembles with Gaussian weights and biases for general nonlinear activation functions, we find that the output distribution only depends on the scalar products between different inputs. We therefore focus on their propagation through a network ensemble. In particular, we study a linear transition operator that advances the signal distribution layer-wise. We conjecture that the spectral properties of this operator can be more informative of trainability than the average spectral density of the input-output Jacobian. Additionally, the distribution of the cosine similarity indicates how well an initialized network can distinguish different inputs.

We further discuss when network layers of finite width are well represented by mean field analysis and when they are not. Furthermore, we highlight important differences in the analysis. By specializing our derivations to ReLUs, we find variants of the He initialization [7] that fulfill the same criteria but also suffer from the same lack of dynamical isometry [14]. In consequence, such initialized neural networks cannot be trained effectively without batch normalization for high depth. To overcome this problem, we propose a simple initialization scheme for ReLU layers that guarantees perfect dynamical isometry. A subset of the weights can still be drawn from Gaussian distributions or chosen as orthogonal while the remaining ones are designed to ensure full signal propagation. Both consistently outperform the He initialization in our experiments on MNIST and CIFAR-10.

## 2 Signal propagation through Gaussian neural network ensembles

### 2.1 Background and notation

We study fully-connected neural network ensembles with zero mean Gaussian weights and biases. We thus make the following assumption:

An ensemble $\{G\}_{L,N_l,\phi,\sigma_w,\sigma_b}$ of fully-connected feed forward neural networks consists of networks with depths $L$, widths $N_l$, $l = 0, ..., L$, independently normally distributed weights and biases with $w_{ij}^{(l)} \sim \mathcal{N}\left(0, \sigma_{w,l}^2\right), b_i^{(l)} \sim \mathcal{N}\left(0, \sigma_{b,l}^2\right)$, and non-decreasing activation function $\phi : \mathbb{R} \to \mathbb{R}$. Starting from the input vector $\mathbf{x}^{(0)}$, signal $\mathbf{x}^{(l)}$ propagates through the network, as usual, as:

$$\mathbf{x}^{(l)} = \phi\left(\mathbf{h}^{(l)}\right), \qquad\qquad \mathbf{h}^{(l)} = \mathbf{W}^{(l)}\mathbf{x}^{(l-1)} + \mathbf{b}^{(l)},$$

$$x_i^{(l)} = \phi\left(h_i^{(l)}\right), \qquad\qquad h_i^{(l)} = \sum_{j=1}^{N_{l-1}} w_{ij}^{(l)} x_j^{(l-1)} + b_i^{(l)},$$

for $l = 1, \ldots, L$, where $\mathbf{h}^{(l)}$ is the pre-activation at layer $l$, $\mathbf{W}^{(l)}$ is the weight matrix, and $\mathbf{b}^{(l)}$ is the bias vector. If not indicated otherwise, 1-dimensional functions applied to vectors are applied to each component separately. To ease notation, we follow the convention to suppress the superscript $(l)$ and write, for instance, $x_i$ instead of $x_i^{(l)}$, $\underline{x}_i$ instead of $x_i^{(l-1)}$, and $\overline{x}_i$ instead of $x_i^{(l+1)}$, when the layer reference is clear from the context.

Ideally, the initialized network is close to the trained one with high probability and can be reached fast in a small number of training steps. Hence, our first goal is to understand the ensemble above and the trainability of an initialized network without requiring mean field approximations of infinite $N_l$. In particular, we derive the probability distribution of the output $\mathbf{x}^{(L)}$. Within this framework, our second goal is to learn how to improve on the He initialization, i.e., the choice $\sigma_{w,l} = \sqrt{2/N_l}$ and $b_i^{(l)} = 0$. Even though it preserves the variance for ReLUs, i.e., $\phi(x) = \max\{0, x\}$, as activation

functions [7], neither this parameter choice nor orthogonal weights lead to dynamical isometry [14]. Thus, the average spectrum of the input-output Jacobian is not concentrated around 1 for higher depths and infinite width. In consequence, ReLUs are argued to be an inferior choice compared to sigmoids [14]. Thus, our third goal is to provide an initialization scheme for ReLUs that overcomes the resulting problems and provides dynamical isometry.

We start with our results about the signal propagation for general activation functions. The proofs for all theorems are given in the supplementary material. As we show, the signal output distribution depends on the input distribution only via scalar products of the inputs. Higher order terms do not propagate through a network ensemble at initialization. In consequence, we can focus on the distribution of such scalar products later on to derive meaningful criteria for the trainability of initialized deep neural networks.

## 2.2 General activation functions

Let's first assume that the signal $\underline{\mathbf{x}}$ of the previous layer is given. Then, each pre-activation component $h_i$ of the current layer is normally distributed as $h_i = \sum_{j=1}^{N_l} w_{ij}\underline{x}_j + b_i \sim \mathcal{N}\left(0, \sigma_w^2 \sum_j \underline{x}_j^2 + \sigma_b^2\right)$, since the weights and bias are independently normally distributed with zero mean. The non-linear monotonically increasing transformation $x_i = \phi(h_i)$ is distributed as $x_i \sim \Phi\left(\frac{\phi^{-1}(\cdot)}{\sigma}\right)$, where $\phi^{-1}$ denotes the generalized inverse of $\phi$, i.e. $\phi^{-1}(x) := \inf\{y \in \mathbb{R} \mid \phi(y) \geq x\}$, $\Phi$ the cumulative distribution function (cdf) of a standard normal random variable, and $\sigma^2 = \sigma_w^2 |\underline{\mathbf{x}}|^2 + \sigma_b^2$. Thus, we only need to know the distribution of $|\underline{\mathbf{x}}|^2$ as input to compute the distribution of $x_i$. The signal propagation is thus reduced to a 1-dimensional problem. Note that the assumption of equal $\sigma_w^2$ for all incoming edges into a neuron are crucial for this result. Otherwise, $h_i \sim \mathcal{N}\left(0, \sum_j \sigma_{w,j}^2 \underline{x}_j^2 + \sigma_{b,i}^2\right)$ would require the knowledge of the distribution of $\sum_j \sigma_{w,j}^2 \underline{x}_j^2$, which depends on the parameters $\sigma_{w,j}^2$. Based on $\sigma_{w,j}^2 = \sigma_w^2$ however, we can compute the probability distribution of outputs.

**Proposition 1.** *Let the probability density $p_0(z)$ of the squared input norm $|\mathbf{x}^{(0)}|^2 = \sum_{i=1}^{N_0}\left(x_i^{(0)}\right)^2$ be known. Then, the distribution $p_l(z)$ of the squared signal norm $|\mathbf{x}^{(l)}|^2$ depends only on the distribution of the previous layer $p_{l-1}(z)$ as transformed by a linear operator $T_l : L^1(\mathbb{R}_+) \to L^1(\mathbb{R}_+)$ so that $p_l = T_l(p_{l-1})$. $T_l$ is defined as*

$$T_l(p)[z] = \int_0^\infty k_l(y,z)p(y)\,dy, \tag{1}$$

*where $k(y,z)$ is the distribution of the squared signal $z$ at layer $l$ given the squared signal at the previous layer $y$ so that $k_l(y,z) = p_{\phi(h_y)^2}^{*N_{l-1}}(z)$, where $*$ stands for convolution and $p_{\phi(h_y)^2}(z)$ denotes the distribution of the squared transformed pre-activation $h_y$, which is normally distributed as $h_y \sim \mathcal{N}\left(0, \sigma_w^2 y^2 + \sigma_b^2\right)$. This distribution serves to compute the cumulative distribution function (cdf) of each signal component $x_i^{(l)}$ as*

$$F_{x_i^{(l)}}(x) = \int_0^\infty dz\, p_{l-1}(z)\Phi\left(\frac{\phi^{-1}(x)}{\sqrt{\sigma_w^2 z + \sigma_b^2}}\right), \tag{2}$$

*where $\phi^{-1}$ denotes the generalized inverse of $\phi$ and $\Phi$ the cdf of a standard normal random variable. Accordingly, the components are jointly distributed as*

$$F_{x_1^{(l)},\ldots,x_{N_l}^{(l)}}(x) = \int_0^\infty dz\, p_{l-1}(z)\Pi_{i=1}^{N_l}\Phi\left(\frac{\phi^{-1}(x_i)}{\sigma_z}\right), \tag{3}$$

*where we use the abbreviation $\sigma_z = \sqrt{\sigma_w^2 z + \sigma_b^2}$.*

As common, the $N$-fold convolution of a function $f \in L^1(\mathbb{R}_+)$ is defined as repeated convolution with $f$, i.e., by induction, $f^{*N}(z) = f * f^{*(N-1)}(z) = \int_0^z f(x)f^{*(N-1)}(z-x)\,dx$. In Prop. 1, we note the radial symmetry of the output distribution. It only depends on the squared norm of the input. For a single input $\mathbf{x}^{(0)}$, $p_0(z)$ is given by the indicator function $p_0(z) = \mathbb{1}_{|\mathbf{x}^{(0)}|^2}(z)$. Interestingly,

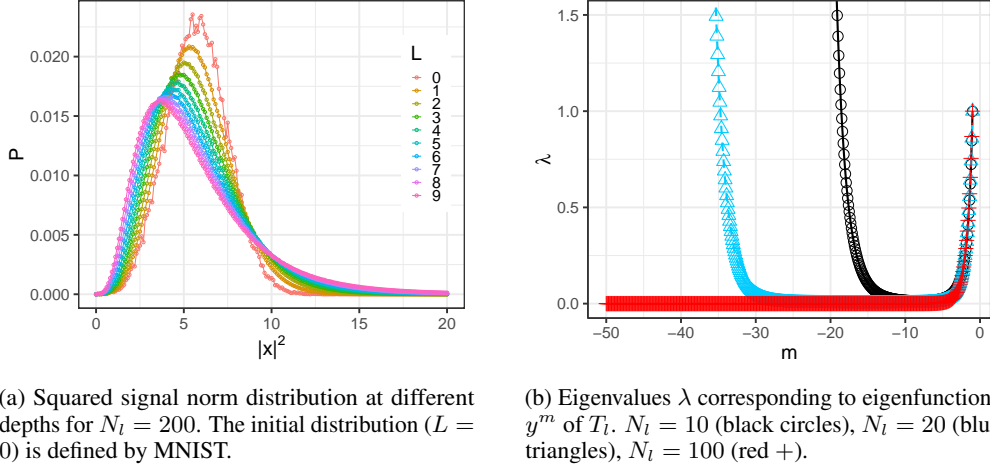

(a) Squared signal norm distribution at different depths for $N_l = 200$. The initial distribution ($L = 0$) is defined by MNIST.

(b) Eigenvalues $\lambda$ corresponding to eigenfunctions $y^m$ of $T_l$. $N_l = 10$ (black circles), $N_l = 20$ (blue triangles), $N_l = 100$ (red +).

Figure 1: Layer-wise transition of the squared signal norm distribution for ReLUs with He initialization parameters $\sigma_w = \sqrt{2/N_l}$, $\sigma_b = 0$.

mean field analysis also focuses on the average or the squared signal, which is likewise updated layer-wise. Prop. 1 explains and justifies the focus of mean field theory on the squared signal norm. More information is not transmitted from layer to layer to determine the state (distribution) of a single neuron. The difference to mean field theory here is that we regard the full distribution $p_{l-1}$ of the previous layer instead of its average only on infinitely large layers. The linear operator $T_l$ governs this distribution. $p_{\mathbf{x}^{(L)}} = \prod_{l=1}^{L} T_l p_{\mathbf{x}^{(0)}}$, where the product is defined by function composition. Hence, the linear operator $\prod_{l=1}^{L} T_l$ can also be interpreted as the Jacobian corresponding to the (linear) function that maps the squared input norm distribution to the squared output norm distribution. $T_l$ is different from the signal input output Jacobian studied in mean field random matrix theory, yet, its spectral properties can also inform us about the trainability of the network ensemble. Conveniently, we only have to study one spectrum and not a distribution of eigenvalues that are potentially coupled as in random matrix theory. For any nonlinear activation function, $T_l$ can be approximated numerically on an equidistant grid. The convolution in the kernel definition can be computed efficiently with the help of Fast Fourier Transformations. The eigenvalues of the matrix approximating $T_l$ define the approximate signal propagation along the eigendirections.

However, we only receive the full picture when we extend our study to look at the joint output distribution, i.e., the outputs corresponding to different inputs.

**Proposition 2.** *The same component of pre-activations of signals* $h_1, ..., h_D$ *corresponding to different inputs* $\mathbf{x}_1^{(0)}, ..., \mathbf{x}_D^{(0)}$, *are jointly normally distributed with zero mean and covariance matrix* $V$ *defined by*

$$v_{ij} = Cov(h_i, h_j) = \sigma_w^2 \langle \underline{\mathbf{x}}_i, \underline{\mathbf{x}}_j \rangle + \sigma_b^2 \tag{4}$$

*for* $i, j = 1, ..., D$ *conditional on the signals* $\underline{\mathbf{x}}_i$ *of the previous layer corresponding to* $\mathbf{x}_i^{(0)}$, *where* $D$ *denotes the number of data points.*

After non-linear activation, the signals are not jointly normally distributed anymore. But their distribution is a function of the squared norms and scalar products between signals of the previous layer only. Thus, it is sufficient to propagate the joint distribution of three variables that can attain different values, i.e., $|\mathbf{x}_1|^2$, $|\mathbf{x}_2|^2$, $\langle \mathbf{x}_1, \mathbf{x}_2 \rangle$, through the layers to determine the joint output distribution of two signals $\mathbf{x}_1$ and $\mathbf{x}_1$ corresponding to different inputs. No other information about the joint distribution of inputs, e.g., higher moments, can influence the ensemble output distribution and thus our choice of weight and bias parameters. In consequence, the focus on quantities corresponding to the above in mean field theory is justified for Gaussian parameter initialization and does not require any approximation. Yet, the mean field assumption that pre-activation signals are exactly normally distributed and not only conditional on the previous signal is approximate. Accordingly, the output distribution for finite neural networks does not follow a Gaussian process with average covariance matrix $V$ as in mean field theory [11]. $V$ follows a probability distribution that is determined by

the previous layers. For the initialization scheme for ReLUs that we propose later, we can state the distribution of $V$ explicitly. First however, we analyze ReLUs in the standard framework and specialize the above theorems.

## 2.3 Rectified Linear Units (ReLUs)

The minimum initialization criterion to avoid vanishing or exploding gradients is to preserve the expected squared signal norm. For finite networks, this is given as follows.

**Corollary 3.** *For ReLUs, the expectation value of the squared signal conditional on the squared signal of the previous layer is given by:*

$$\mathbb{E}\left(|\mathbf{x}^{(l)}|^2 \mid |\mathbf{x}^{(l-1)}|^2 = \underline{y}\right) = (\sigma_{w,l}^2 \underline{y} + \sigma_{b,l}^2)\frac{N_l}{2}. \tag{5}$$

*Consequently, the expectation of the final squared signal norm depends on the initial input as:*

$$\mathbb{E}\left(|\mathbf{x}^{(L)}|^2 \big| |\mathbf{x}^{(0)}|^2\right) = |\mathbf{x}^{(0)}|^2 \Pi_{l=1}^L \frac{N_l \sigma_{w,l}^2}{2} + \sigma_{b,L}^2 \frac{N_L}{2} + \sum_{l=1}^{L-1} \sigma_{b,l}^2 \frac{N_l}{2} \Pi_{n=l+1}^L \frac{N_n \sigma_{w,n}^2}{2} \tag{6}$$

Similar relations for the expected signal components and their variance follow from Eq. (6) and are covered in the supplementary material. [6] has derived a simpler version of Eq. (6) for equal $\sigma_{w,l}$ and $N_l$ across layers.

A straightforward way to preserve the average squared signal or the squared output signal norm distribution is exactly the He initialization $\sigma_{b,l} = 0$ and $\sigma_{w,l} = \sqrt{2/N_l}$ [7], which is also confirmed by mean field analysis. Yet, we have many more choices even when $\sigma_{b,l}^2 = 0$. We only need to fulfill one condition, i.e., $0.5^L \Pi_{l=1}^L N_l \sigma_{w,l}^2 \approx 1$. In case that we normalize the input so that $|\mathbf{x}^{(0)}|^2 = 1$, $\sigma_{b,l}^2 \neq 0$ is also a valid option and we have $2L - 1$ degrees of freedom.

There remains the question whether there exist further criteria to be fulfilled that improve the trainability of the initial network ensemble. The whole output distribution could provide those and its derivation is given in the supplementary material. According to Prop. 2, it is guided by the layer-wise joint distribution of the variables $\left(|\mathbf{x}_1|^2, |\mathbf{x}_2|^2, \langle\mathbf{x}_1, \mathbf{x}_2\rangle\right)$ given $\left(|\underline{\mathbf{x}}_1|^2, |\underline{\mathbf{x}}_2|^2, \langle\underline{\mathbf{x}}_1, \underline{\mathbf{x}}_2\rangle\right)$. As this is computationally intensive to obtain, we focus on marginals, i.e., the distributions of $|\mathbf{x}|^2$ and $\langle\mathbf{x}_1, \mathbf{x}_2\rangle$. These are sufficient to highlight several drawbacks of the initialization approach and provide us with insights to propose an alternative that overcomes these shortcomings.

First, we focus on $|\mathbf{x}^{(l)}|^2$ and derive a closed form solution for the integral kernel $k_l(y, z)$ of $T_l$ in Prop. 1 and analyse some of its spectral properties for ReLUs. This allows us to reason about the shape of the stationary distribution of $T_l$, i.e., the limit output distribution for networks with increasing depth.

**Proposition 4.** *For ReLUs, the linear operator $T_l$ in Prop. 1 is defined by*

$$k_l(y, z) = 0.5^{N_l}\left(\delta_0(z) + \sum_{k=1}^{N_l}\binom{N_l}{k}\frac{1}{\sigma_y^2}p_{\chi_k^2}\left(\frac{z}{\sigma_y^2}\right)\right) \tag{7}$$

*with $\sigma_y = \sqrt{\sigma_w^2 y + \sigma_b^2}$, where $\delta_0(z)$ denotes the $\delta$-distribution peaked at $0$ and $p_{\chi_k^2}$ the density of the $\chi^2$ distribution with $k$ degrees of freedom. For $\sigma_b = 0$, the functions $f_m(y) = y^m \mathbb{1}_{]0,\infty]}(y)$ are eigenfunctions of $T_l$ for any $m \in \mathbb{R}$ (even though they are not elements of $L^1(\mathbb{R}_+)$ and thus not normalizable as probability measures) with corresponding eigenvalue $\lambda_{l,m} \in \mathbb{R}$: $T_l f_m = \lambda_{l,m} f_m$ with*

$$\lambda_{l,m} = 0.5^{N_l - m - 1}\frac{1}{\sigma_w^{2m+2}}\sum_{k=1}^{N_l}\binom{N_l}{k}\frac{\Gamma(k/2 - m - 1)}{\Gamma(k/2)} \tag{8}$$

Note that, for $\sigma_b = 0$, the eigenfunctions $y^m$ cannot be normalized on $\mathbb{R}_+$, as the antiderivative diverges at zero for $m \leq -1$. However, if we discretize $T_l$ in numerical experiments they can be normalized and the real eigenvectors representing probability distributions attain shapes $\approx y^m$.

Fig. 1a provides an example of the output distribution for 9 layers each consisting of $N_l = 200$ neurons with He initialization parameters. The average squared signal is indeed preserved but becomes more right tailed for deeper layers. Fig. 1b shows the corresponding eigenvalues of $T_l$ as in Prop. 4. In summary, we observe a window $m_{\text{crit}} < m \leq -1$ with eigenvalues $\lambda_{l,m} < 1$. Specifically, for the He values $\sigma_{b,l} = 0$ and $\sigma_{w,l} = \sqrt{2/N_l}$, numerical experiments reveal a relation $m_{\text{crit}} \approx -3.2559793 - 1.6207083 N_l$. Signal parts within this window are damped down in deeper layers, while the remaining parts explode. Only $y^{m_{\text{crit}}}$ is preserved through the layers and depends on the choice of $\sigma_{w,l}$. Interestingly, for $m = -1$, $\lambda_{l,m}$ is independent of $\sigma_{w,l}$ and given by $\lambda_{l,-1} = 1 - 0.5^{N_l}$. Thus, it approaches $\lambda_{l,-1} = 1$ for increasing $N_l$. For the He initialization, $y^{m_{\text{crit}}}$ converges to the $\delta_0(y)$ for increasing $N_l$. In contrast to mean field analysis, not the whole space of eigenfunctions corresponds to eigenvalue 1 for the He initialization. In particular, eigenvalues bigger than one exist that can be problematic for exploding gradients. To reduce their number, broader layers promise better protection as well as smaller values of $\sigma_{w,l}$. Ultimately, we care about the product of layer-wise eigenvalues, i.e., the eigenvalues of $\Pi_l T_l$. Again, setting these to 1 imposes a constraint only on the product $\Pi_l \sigma_{w,l}^2$ like in Cor. 3. Hence, we gain no additional constraint on our initial parameters and have no means to prevent eigenvalues larger than 1.

The biggest challenge for trainability, however, is the ability to differentiate similar signals. We therefore study the evolution of the cosine similarity $\langle \mathbf{x}_1^{(l)}, \mathbf{x}_2^{(l)} \rangle$ of two inputs $\mathbf{x}_1^{(0)}$ and $\mathbf{x}_2^{(0)}$ or the unnormalized scalar product through layers $l$.

**Theorem 5.** *For ReLUs, let $x_1 = \phi(h_1), x_2 = \phi(h_2)$ be the same signal component, i.e., neuron, where each corresponds to a different input $\mathbf{x}_1^{(0)}$ or $\mathbf{x}_2^{(0)}$. Let the correlation $\rho = v_{12}/\sqrt{v_{11}v_{22}}$ of the pre-activations $h_1, h_2$ be given, where $V$ denotes the 2-dimensional covariance matrix as defined in Prop. 2. Then, the correlation after non-linear activation is*

$$Cor(x_1, x_2) = \frac{\sqrt{1-\rho^2} - 1 + 2\pi\rho g(\rho)}{\pi - 1}. \tag{9}$$

*$g(\rho)$ is defined as $g(\rho) = \frac{1}{\sqrt{2\pi}} \int_0^\infty \Phi\left(\frac{\rho}{\sqrt{1-\rho^2}} u\right) \exp\left(-\frac{1}{2}u^2\right) du$ for $|\rho| \neq 1$ and $g(-1) = 0$, $g(1) = 0.5$. The average of the sum of all components $\mathbb{E}\left(\langle \mathbf{x}_1, \mathbf{x}_2 \rangle\right)$ conditional on the previous layer is:*

$$\mathbb{E}\left(\langle \mathbf{x}_1, \mathbf{x}_2 \rangle \mid \rho\right) = N_l \sqrt{v_{11}v_{22}} \left(g(\rho)\rho + \frac{\sqrt{1-\rho^2}}{2\pi}\right) \approx N_l \sqrt{v_{11}v_{22}} \frac{1}{4}(\rho + 1). \tag{10}$$

*Furthermore, conditional on the signals of the previous layer, $\langle \mathbf{x}_1, \mathbf{x}_2 \rangle$ is distributed as $f_{\text{prod}}^{*N_l}(t)$, where $f_{\text{prod}}(y) = (1 - g(\rho))\delta_0(y) + \frac{1}{2\pi\sqrt{\det(V)}} \exp\left(\frac{v_{12}y}{2\det(V)}\right) K_0\left(\frac{\sqrt{v_{11}v_{22}}}{\det(V)} y\right)$ and $K_0(w) = \int_0^\infty \cos(w \sinh(t)) dt$ denotes the modified Bessel function of second kind.*

Note that [2] studies a similar integral but in the mean field limit. The correlation of the signal components only depends on $\rho$ (and is always smaller than $\rho$). Analogous to the c-map in mean field approaches [18], the actual quantity of interest would be the distribution of the correlation $\rho$, i.e., $\rho = \frac{\sigma_w^2 \langle \mathbf{x}_1, \mathbf{x}_2 \rangle + \sigma_b^2}{\sqrt{\left(\sigma_w^2 |\mathbf{x}_1|^2 + \sigma_b^2\right)\left(\sigma_w^2 |\mathbf{x}_2|^2 + \sigma_b^2\right)}}$. Interestingly, for $\sigma_b = 0$, $\rho = \frac{\langle \mathbf{x}_1, \mathbf{x}_2 \rangle}{|\mathbf{x}_1||\mathbf{x}_2|}$ coincides with the cosine similarity of the two signals. The preservation of this quantity on average has been shown to be the most indicative criterion for trainability of ReLU residual networks [24]. We therefore take a closer look at its distribution. To save computational time and space, we sample $N_l$ components iid from a 2 dimensional normal distribution as introduced in Prop. 2 and transform the components by ReLUs to obtain two vectors $\mathbf{x}_1$ and $\mathbf{x}_2$ and calculate their cosine similarity. Repeating this procedure $10^6$ times results in Fig. 2. First, we note that correlations can only be positive after the first layer, since all signal components are positive (or zero) after transformation by ReLUs. Negative cosine similarities cannot be propagated through Gaussian ReLU ensembles. Data transformation to obtain positive inputs can mitigate this issue. Yet, Eq. (10) highlights an unavoidable problem for deep models, i.e., the average cosine similarity increases from layer to layer until it reaches 1 at high depths. Then, all signals become parallel and thus indistinguishable.

While this effect cannot be mitigated completely within our initialization scheme, a slightly smaller choice of $\sigma_w$ than the He initialization reduces the average cosine distance and a smaller number

of neurons in one layer increases the variance of the cosine distance, as shown in Fig. 2. A higher variance increases the probability that smaller values of the cosine distance can be propagated. We hypothesize that this effect contributes to the better trainability of ReLU neural networks with DropOut or DropConnect [22], since both reduce the effective number of neurons $N_l$. Yet, a smaller $\sigma_w$ and DropOut (or DropConnect) introduce a risk of vanishing gradients in deep neural networks [17]. An adjustment of $\sigma_w$ by the DropOut rate to avoid this effect [17] would also destroy possible beneficial effects on the cosine similarity.

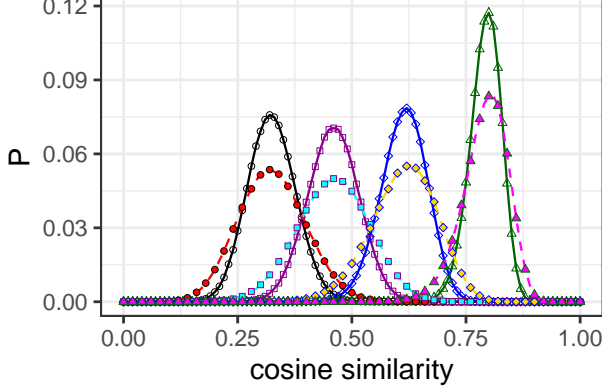

Figure 2: Probability distribution of the cosine similarity conditional on the previous layer with $|\underline{\mathbf{x}}_1| = |\underline{\mathbf{x}}_2| = 1$ and $\langle \underline{\mathbf{x}}_1, \underline{\mathbf{x}}_2 \rangle$ equals 0 (circles), 0.25 (squares), 0.5 (diamonds), and 0.75 (triangles) for $N_l = 100$ (dashed lines and filled symbols) and $N_l = 200$ (lines and unfilled symbols) neurons.

For smaller $\sigma^2 = 1/N$, [1] observes a phenomenon related to the cosine similarity, i.e. shattered gradients. However, in this setting, the effect of vanishing gradients and increasing correlations are indistinguishable. In fact, the authors observe decreasing correlations, while we highlight the problem of increasing ones for the He initialization. Interestingly, in the He case ($\sigma^2 = 2/N$), [1] finds that also batch normalization cannot provide better trainability. For "typical" inputs that are shown to be common in networks with batch normalization (but not in networks without, which we study here), the covariance between outputs corresponding to different inputs decays exponentially.

We therefore discuss an alternative solution that [1] proposes also for convolutional and residual neural networks and has first been introduced by [21].

## 3 Alternative initialization of fully-connected ReLU deep neural networks

The issues of training deep neural networks with ReLUs are caused by the fact that negative signal can never propagate through the network and a neuron's state is zero in half of the cases. Hence, we discuss an initialization, where the full signal is transmitted. We set the bias vector $b_i^{(l)} = 0$ and the weight matrices $W^{(l)} \in \mathbb{R}^{N_{l-1} \times N_l}$ are initially determined by a submatrix $W_0^{(l)} \in \mathbb{R}^{\frac{N_{l-1}}{2} \times \frac{N_l}{2}}$ as

$$W^{(l)} = \left[ \begin{array}{cc} W_0^{(l)} & -W_0^{(l)} \\ -W_0^{(l)} & W_0^{(l)} \end{array} \right].$$

Regardless of the choice of $W_0^{(l)}$, we receive a signal vector $\mathbf{x}^{(l)}$, where half of the entries correspond to the positive part of the pre-activations and the second half to the negative part, i.e., if $i \le N_l/2$ and $h_i^{(l)} = \sum_j w_{ij}^{(l)} x_j^{(l-1)} > 0$, we have $x_i^{(l)} = h_i^{(l)}$ and $x_{i+N_l/2}^{(l)} = 0$ or the other way around for $h_i^{(l)} < 0$. This way, effectively $h_i^{(l)} = \sum_{j=1}^{N_l/2} w_{0,ij}^{(l)} h_j^{(l-1)}$ is propagated so that we have initially linear networks $\mathbf{h}^{(L)} = \prod_{l=0}^{L} W_0^{(l)} \mathbf{h}_0$. Note that we still have to train the full $N_{l-1} N_l$ parameters of $W^{(l)}$ and can learn non-linear functions. [21] observed that convolutional neural networks even trained the first layers to resemble linear neural networks, which inspired this choice of initialization.

In this setting, we have several good choices of $W_0^{(l)}$. First, we assume iid entries $w_{0,ij}^{(l)} \sim \mathcal{N}\left(\mathbf{0}, \sigma_{w,l}^2\right)$ as before. We call this variant *Gaussian submatrix* (GSM) initialization. In this case, our assumptions from the previous sections are met and we can repeat the analysis for networks of width $N_l/2$ and $\phi(h) = h$, i.e. set the activation function to the identity. The same parameter choice as in Cor. 3, e.g., $\sigma_{w,l}^2 = 2/N_l$, preserves the variance and now also the cosine distance between signals corresponding to different inputs. The analysis is rather straight-forward and the output distribution is defined by the distribution of $\prod_{l=0}^{L} W_0^{(l)} \mathbf{x}^{(\mathbf{0})}$. $\prod_{l=0}^{L} W_0^{(l)}$ follows a product Wishart distribution with known spectrum [13, 14].

According to [14] however, dynamical isometry leads to better training results and speed. This demands an input-output Jacobian close to the identity or a spectrum of the signal propagation operator $T_l$ that is highly concentrated around 1. Previously, this was believed to be better achievable by $tanh$ or sigmoid rather than by ReLU as choice of activation functions [14]. Yet, in the parameter sharing setting above, perfect dynamical isometry for $h$ can be achieved by orthogonal $W_0^{(l)}$, i.e., it is drawn uniformly at random from all matrices $W$ fulfilling $W^T W = \sigma_{w,l}^2 I$ with $\sigma_{w,l}^2 = 1$. This is our second initialization proposal in addition to GSM.

Alternatively, [25] recommends to shift the signal $h_i^{(l)}$ by a non-zero bias $b_i^{(l)}$ to enable negative signal to pass through a ReLU activation instead of the proposed parameter sharing solution. We also considered a similar approach but decided for the proposed one as it is point-symmetric, guaranties therefore perfect dynamical isometry, is computational less intensive, as it does not need to compute a data dependent bias $b_i^{(l)}$ as in batch normalization, and is more convenient for theoretical analysis, which can rely on a rich literature on linear deep neural networks.

## 4 Experiments for different initialization schemes

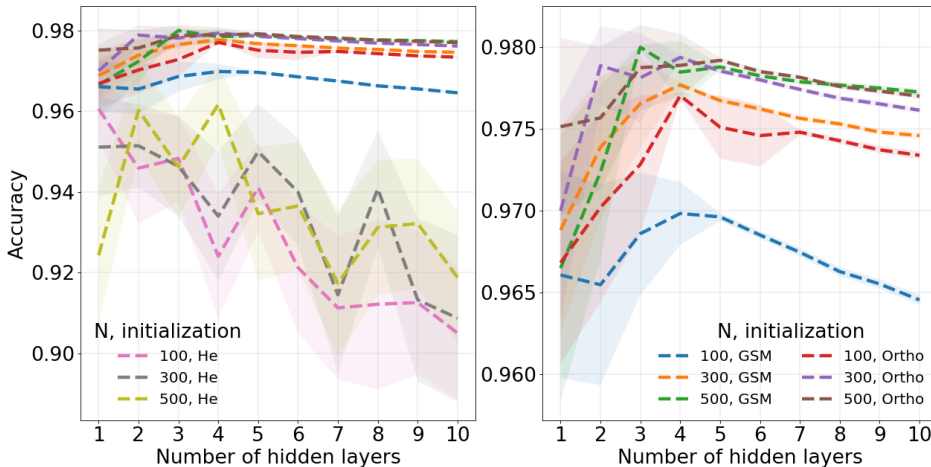

Figure 3: Classification test accuracy on MNIST for different widths $N$, depths $L$, and weight initialization with parameters $\sigma_b = 0$, $\sigma_w = \sqrt{2/N}$ for He and GSM initialization, and $\sigma_w = 1$ for orthogonal $W_0$ after $10^4$ SGD steps. We report the average of 100 realizations and the corresponding 0.95 confidence interval. The right plot is a section of the left. Note that the legends apply to both plots.

We train fully-connected ReLU feed forward networks of different depth consisting of $L = 1, \ldots, 10$ hidden layers with the same number of neurons $N_l = N = 100, 300, 500$ and an additional softmax classification layer on MNIST [10] and CIFAR-10 [9] to compare three different initialization schemes: the standard He initialization and our two proposals in Sec. 3, i.e., GSM and orthogonal weights. We focus on minimizing the cross-entropy by Stochastic Gradient Descent (SGD) without batch normalization or any data augmentation techniques. Hence, our goal is not to outperform the classification state of the art but to compare the initialization schemes under similar realistic conditions. Since deep networks normally require a smaller learning rate than the ones with a small number of hidden layers, as in Ref. [14], we adapt the learning rate to $(0.0001 + 0.003 \cdot \exp(-step/10^4))/L$ for MNIST and $(0.00001 + 0.0005 \cdot \exp(-step/10^4))/L$ for CIFAR-10 for $10^4$ SGD steps with a batch size of 100 in all cases. To reduce the number of parameters and speed up computations, we clipped original CIFAR-10 images to $28 \times 28$ size. For each configuration, we train 100 instances on MNIST and 30 instances on CIFAR-10 and report the average accuracy with a 0.95 confidence interval in Fig. 3 and Fig. 4 respectively. Each experiment on MNIST was run on 1 Nvidia GTX 1080 Ti GPU, while each experiment on CIFAR-10 was performed on 4 Nvidia GTX 1080 Ti GPUs.

Note that the accuracy on CIFAR-10 is lower than for convolutional architectures, as we restrict ourselves to deep fully-connected networks to focus on their trainability. [1] shows that a similar

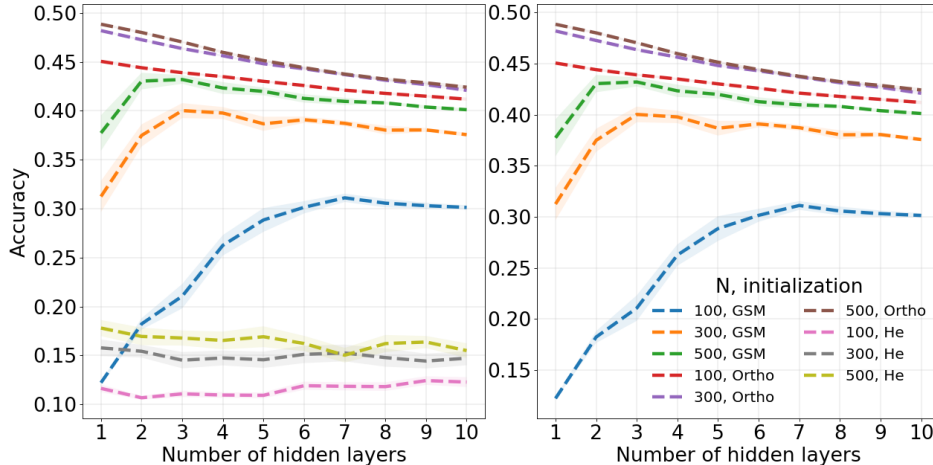

Figure 4: Classification test accuracy on CIFAR-10 for different widths $N$, depths $L$, and weight initialization with parameters $\sigma_b = 0$, $\sigma_w = \sqrt{2/N}$ for He and GSM initialization, and $\sigma_w = 1$ for orthogonal $W_0$ after $10^4$ SGD steps. We report the average of 30 realizations and the corresponding 0.95 confidence interval. The right plot is a section of the left.

orthogonal initialization improves training results also for convolutional and residual neural networks. As suggested by our theoretical analysis, both proposed initialization schemes consistently outperform the He initialization and show stable training results, in particular, for deeper network architectures, where the He initialized networks decrease in accuracy. GSM and orthogonal $W_0$ both perform better for higher width $N$, while orthogonal $W_0$ seems to be the most reliable choice.

## 5   Discussion

We have introduced a framework for the analysis of deep fully-connected feed forward neural networks at initialization with zero mean normally distributed weights and biases. It is exact, does not rely on mean field approximations, provides distribution information of output and joint output signals, and applies to networks with arbitrary layer widths. It has led to the insight that only the scalar products between inputs determine the shape of the output distribution, but it is not influenced by higher interaction terms.

Hence, for ReLUs, we have analysed the propagation of these quantities through the deep neural network ensemble. While mean field analysis provides only the He initialization for good training results, we have extended the number of possible parameter choices that avoid vanishing or exploding gradients. However, no parameter choice can avoid the tendency that signals become more aligned with increasing depth. Deep ReLU Gaussian neural network ensembles cannot distinguish different input correlations and are therefore not well trainable without batch normalization. Even batch normalization does not guaranty the transmission of correlations between different inputs.

As solution to this problem, we have discussed an alternative but simple initialization scheme that relies on initial parameter sharing. One variant guarantees perfect dynamical isometry. Experiments on MNIST and CIFAR-10 demonstrate that deeper fully-connected ReLU networks can become better trainable in the proposed way than by the standard approach.

## Acknowledgments

We would like to thank Joachim M. Buhmann and Alkis Gotovos for their valuable feedback on the manuscript and the reviewers for their insightful comments. This work was partially funded by the Swiss Heart Failure Network, PHRT122/2018DRI14 (J. M. Buhmann, PI). RB was supported by a grant from the US National Cancer Institute (1R35CA220523).

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
