[Supplementary Material]

# Supplementary material: Initialization of ReLUs for dynamical isometry

**Rebekka Burkholz**
Department of Biostatistics
Harvard T.H. Chan School of Public Health
655 Huntington Avenue, Boston, MA 02115
rburkholz@hsph.harvard.edu

**Alina Dubatovka**
Department of Computer Science
ETH Zurich
Universitätstrasse 6, 8092 Zurich
alina.dubatovka@inf.ethz.ch

## 1 Proofs of theorems

**Proposition 1.** *We assume that the probability density $p_0(z)$ of the squared input $|\mathbf{x}^{(0)}|^2 = \sum_{i=1}^{N_0} x_i^{(0)}$ is known. Then, the distribution $p_l(z)$ of the squared signal vector $|\mathbf{x}^{(l)}|^2$ depends only on the distribution of the previous layer as transformation by a linear operator $T_l : L^1(\mathbb{R}_+) \to L^1(\mathbb{R}_+)$ so that $p_l = T_l(p_{l-1})$. $T$ is defined as*

$$T_l(p)[z] = \int_0^\infty k_l(y,z)p(y)dy, \tag{1}$$

*where $k(y,z)$ is the distribution of the squared signal $z$ at layer $l$ given the squared signal at the previous layer $y$ so that $k_l(y,z) = p_{\phi(h_y)^2}^{*N_{l-1}}(z)$, where $*$ stands for convolution and $p_{\phi(h_y)^2}(z)$ denotes the distribution of the squared transformed pre-activation $h_y$, which is normally distributed as $h_y \sim \mathcal{N}\left(0, \sigma_w^2 y^2 + \sigma_b^2\right)$. This distribution serves to compute the cumulative distribution function (cdf) of each signal component $x_i^l$ as*

$$F_{x^{(l)}}(x) = \int_0^\infty dz p_{l-1}(z)\Phi\left(\frac{\phi^{-1}(x)}{\sqrt{\sigma_w^2 z + \sigma_b^2}}\right), \tag{2}$$

*where $\phi^{-1}$ denotes the generalized inverse of $\phi$ and $\Phi$ the cdf of a standard normal random variable. Accordingly, the components are jointly distributed as*

$$F_{x_1^{(l)},\ldots,x_{N_l}^{(l)}}(x) = \int_0^\infty dz p_{l-1}(z)\Pi_{i=1}^{N_l}\Phi\left(\frac{\phi^{-1}(x_i)}{\sigma_z}\right), \tag{3}$$

*where we use the abbreviation $\sigma_z = \sqrt{\sigma_w^2 z + \sigma_b^2}$.*

*Proof.* Let's focus on a signal vector $\mathbf{x}$ conditional on the signal of the previous layer $\underline{\mathbf{x}}$ . As explained in the main manuscript, a single signal component is distributed as $x_i \sim \Phi\left(\frac{\phi^{-1}(\cdot)}{\sigma_i}\right)$ with $\sigma_i^2 = \sigma_w^2 |\underline{\mathbf{x}}|^2 + \sigma_b^2$ given $|\underline{\mathbf{x}}|^2$. As the weights and biases are independent in the computation of different components, also the components $x_i$ given $|\underline{\mathbf{x}}|^2$ are independent. Their joint distribution is therefore just the product of the marginal distributions. Their distribution depends on the previous signal only via the squared norm $|\underline{\mathbf{x}}|^2$, which is distributed as $p_{l-1}$. According to Bayes' theorem, we only need the knowledge of the distribution $p_{l-1}(\cdot)$ of the squared norm at the previous layer to determine the signal distribution by Equation 3. We receive $p_l$ of the next layer by successive conditioning on the previous layer and the application of Bayes' Theorem that states

$$p_l(y) = \int_0^\infty p_{|\mathbf{x}|^2|\,|\underline{\mathbf{x}}|^2}p_{l-1}(y)\,dy. \tag{4}$$

Equation 1 follows with the definition $k_l(y,z) = p_{|\mathbf{x}|^2 \big| |\underline{\mathbf{x}}|^2 = y}(z)$. Thus, the kernel $k_l(y,z)$ is defined as the density of the random variable $|\mathbf{x}|^2$ given the squared norm $|\underline{\mathbf{x}}|^2$ of the previous layer. We have to deduce its distribution to conclude the proof. $|\mathbf{x}|^2 = \sum_{i=1}^{N_l} x_i^2 = \sum_{i=1}^{N_l} \phi(h_i)^2$, where the random variables $\phi(h)_i^2$ are independent and the pre-activations are normally distributed as $h_i \sim \mathcal{N}\left(0, \sigma_w^2 |\underline{\mathbf{x}}|^2 + \sigma_b^2\right)$ conditional on the previous layer. Thus, the $x_i^2 \sim \Phi(\frac{\phi^{-1}(\sqrt{(\cdot)})}{\sqrt{\sigma_w^2 |\underline{\mathbf{x}}|^2 + \sigma^2}})$ are also identically independently distributed and so that their sum is given by their convolution:

$$|\mathbf{x}|^2 \sim \mathcal{F}^{-1}\left(\mathcal{F}\Phi(\frac{\phi^{-1}(\sqrt{(\cdot)})}{\sqrt{\sigma_w^2 |\underline{\mathbf{x}}|^2 + \sigma^2}})\right)^{*N_l}, \text{ where } \mathcal{F} \text{ denotes the Fourier transformation.} \qquad \square$$

**Proposition 2.** *For rectified linear units, the linear operator $T$ in Theorem 1 is defined by*

$$k_l(y,z) = 0.5^{N_l}\left(\delta_0(z) + \sum_{k=1}^{N_l}\binom{N_l}{k}\frac{1}{\sigma_y^2}p_{\chi_k^2}\left(\frac{z}{\sigma_y^2}\right)\right) \qquad (5)$$

*with $\sigma_y = \sqrt{\sigma_w^2 y + \sigma_b^2}$. For $\sigma_b = 0$, the functions $f_m(y) = y^m \mathbb{1}_{]0,\infty]}(y)$ are eigenfunctions of $T_l$ for any $m \in \mathbb{R}$ (even though they are not elements of $L^1(\mathbb{R}_+)$ and thus not normalizable as probability measures) with corresponding eigenvalue $\lambda_{l,m} \in \mathbb{R}$: $T_l f_m = \lambda_{l,m} f_m$ with*

$$\lambda_{l,m} = 0.5^{N_l - m - 1}\frac{1}{\sigma_w^{2m+2}}\sum_{k=1}^{N_l}\binom{N_l}{k}\frac{\Gamma(k/2 - m - 1)}{\Gamma(k/2)}. \qquad (6)$$

*Proof.* We specialize Prop. 1 to rectified linear units, i.e. $\phi(x) = \max(0,x)$. Thus, the components $x_i = \phi(h_i)$ are the sum of a $\delta$ distribution in 0, i.e. $\delta_0$, and a truncation of a normal distribution with mean 0 and variance $\sigma_y^2$ given the previous layer $|\underline{\mathbf{x}}|^2 = y$.

$x_i \big| |\underline{\mathbf{x}}|^2 = y \sim 0.5\delta_0(\cdot) + \mathbb{1}_{]0,\infty]}(\cdot)p_{\mathcal{N}}\left(\frac{\cdot}{\sigma_y}\right)/\sigma_y$, where $p_{\mathcal{N}}(z) = \exp(-z^2/2)/\sqrt{2\pi}$ denotes the density of a Standard normal random variable. The squared component $x_i^2 = (\phi(h_i))^2 \sim p_s(\cdot) = 0.5\delta_0(\cdot) + 0.5\frac{1}{\sigma_y}p_{\chi^2}(\frac{\cdot}{\sigma_y})$ is thus either 0 with probability 0.5, i.e. the probability $\Phi(0) = 0.5$ that $h_i$ is negative or, in case that $h_i$ is positive with probability 0.5, it follows a $\chi^2$ distribution as a squared normal random variable with mean 0. As the squared components are independent and identically distributed given the previous layer, their sum (given the previous layer) is distributed as the $N_l$th convolution of the squared component distribution, i.e. $p_l \big| |\mathbf{x}|^2 \sim p_s^{*N_l}$. Its functional form is provided by Equation (5), which adds all possible cases where $N_l - k$ of the components are zero, i.e. $x_i^2 = 0$ and the remaining $k$ are positive and $\chi^2$ distributed. As the sum of $k$ independent $\chi^2$ distributed random variables is $\chi_k^2$ distributed, Equation (5) follows.

To show that $T_l f_m = \int_0^\infty k_l(y,z)f_m(y)dy = \lambda_{l,m}f_m(z)$, we focus on the summands $\int_0^\infty \delta_0(y)f_m(y)\,dy$ and $\int_0^\infty \frac{1}{\sigma_y^2}p_{\chi_k^2}\left(\frac{z}{\sigma_y^2}\right)f_m(y)\,dy$ separately. First, we have $\int_0^\infty \delta_0(y)f_m(y)\,dy = f_m(0) = 0$. Second, we note that $\sigma_y^2 = y\sigma_w^2$ for $\sigma_b = 0$ and integrate

$$\int_0^\infty \frac{1}{\sigma_y^2}p_{\chi_k^2}\left(\frac{z}{\sigma_y^2}\right)f_m(y)\,dy = \frac{0.5^{k/2}}{\Gamma(k/2)}\int_0^\infty \frac{z^{k/2-1}}{\sigma_w^k y^{k/2}}$$

$$\times \exp\left(\frac{z}{2\sigma_w^2 y}\right)y^m\,dy = \frac{z^m}{\sigma_w^{2(m+1)}}\frac{0.5^{k/2}}{\Gamma(k/2)}\int_0^\infty \exp\left(x/2\right)$$

$$x^{k/2-(m+1)-1}\,dx = z^m\frac{2^{m+1}}{\sigma_w^{2(m+1)}}\frac{\Gamma(k/2-(m+1))}{\Gamma(k/2)}$$

after use of the definition of the $\chi^2$ density and a variable transformation $x = z/(\sigma_w^2 y)$. Accordingly, the integration of the product with the full kernel $k(y,z)$ leads to

$$T_l f_m = z^m\frac{2^{m+1}}{\sigma_w^{2(m+1)}}0.5^{N_l}\sum_{k=1}^{N_l}\binom{N_l}{k}\frac{\Gamma(k/2-(m+1))}{\Gamma(k/2)}$$

$$= z^m\lambda_{l,m}.$$

$\square$

**Corollary 3.** *For rectified linear units, the expectation value of the squared signal conditional on the squared signal of the previous layer is given by:*

$$\mathbb{E}\left(|\mathbf{x}^{(l)}|^2\,\big|\,|\mathbf{x}^{(l-1)}|^2 = \underline{y}\right) = (\sigma_w^2 \underline{y} + \sigma_b^2)\frac{N_l}{2}. \tag{7}$$

*Consequently, the expectation of the final squared signal norm depends on the initial input as:*

$$
\begin{aligned}
\mathbb{E}\left(|\mathbf{x}^{(L)}|^2\,\big|\,|\mathbf{x}^{(0)}|^2\right) =& |\mathbf{x}^{(0)}|^2 \Pi_{l=1}^{L} \frac{N_l \sigma_{w,l}^2}{2} + \sigma_{b,L}^2 \frac{N_L}{2} \\
&+ \sum_{l=1}^{L-1} \sigma_{b,l}^2 \frac{N_l}{2} \Pi_{n=l+1}^{L} \frac{N_n \sigma_{w,n}^2}{2}
\end{aligned}
\tag{8}
$$

*The expectation value and variance of a single signal component conditional on the squared signal norm of the previous layer are given by:*

$$\mathbb{E}\left(x_i\,\big|\,|\underline{\mathbf{x}}|^2 = \underline{y}\right) = \frac{1}{\sqrt{2\pi}} \sqrt{\sigma_w^2 \underline{y} + \sigma_b^2}, \tag{9}$$

$$\mathbb{V}\left(x_i\,\big|\,|\underline{\mathbf{x}}|^2 = \underline{y}\right) = \frac{\pi - 1}{2\pi}(\sigma_w^2 \underline{y} + \sigma_b^2). \tag{10}$$

*The last layer depends on the input as:*

$$
\begin{aligned}
\mathbb{E}\left(x_i^{(L)}\,\big|\,\mathbf{x}^{(0)}\right) \leq& \frac{1}{\sqrt{2\pi}} \Big[|\mathbf{x}^{(0)}|^2 \Pi_{l=1}^{L-1} \frac{N_l \sigma_{w,l}^2}{2} \sigma_{w,L}^2 \\
&+ \sigma_b^2 + \sigma_{w,L}^2 \big(\sigma_{b,L-1}^2 \frac{N_{L-1}}{2} \\
&+ \sum_{l=1}^{L-1} \sigma_{b,l}^2 \frac{N_l}{2} \Pi_{n=l+1}^{L-1} \frac{N_n \sigma_{w,n}^2}{2}\big)\Big]^{1/2}, \\
\mathbb{V}\left(x_i^{(L)}\,\big|\,\mathbf{x}^{(0)}\right) =& |\mathbf{x}^{(0)}|^2 \frac{\pi - 1}{2\pi} \sigma_{w,L}^2 \Pi_{l=1}^{L-1} \frac{N_l \sigma_{w,l}^2}{2} + \frac{\pi - 1}{2\pi} \\
&\times \big(\sigma_{b,L}^2 + \sigma_{w,L}^2 \big(\sigma_{b,L-1}^2 \frac{N_{L-1}}{2} \\
&+ \sum_{l=1}^{L-1} \sigma_{b,l}^2 \frac{N_l}{2} \Pi_{n=l+1}^{L-1} \frac{N_n \sigma_{w,n}^2}{2}\big)\big).
\end{aligned}
$$

*Proof.* Prop. 2 states that $|\mathbf{x}^{(l)}|^2\,\big|\,|\mathbf{x}^{(l-1)}|^2 = y \sim k_l(y, \cdot)$ as given by Equation (5). Using the linearity of expectation and the knowledge of the average of a $\chi^2$ distribution $\int_0^\infty \frac{z}{\sigma_y^2} p_{\chi_k^2}\left(\frac{z}{\sigma_y^2}\right)\,dz = k\sigma_y^2$, we receive

$$
\begin{aligned}
\mathbb{E}\left(|\mathbf{x}^{(l)}|^2\,\big|\,|\mathbf{x}^{(l-1)}|^2 = y\right) =& 0.5^{N_l} \sum_{k=1}^{N_l} \binom{N_l}{k} \int_0^\infty p_{\chi_k^2}\left(\frac{z}{\sigma_y^2}\right) \\
\times \frac{z}{\sigma_y^2}\,dz =& 0.5^{N_l} \sigma_y^2 \sum_{k=1}^{N_l} \binom{N_l}{k} k = 0.5^{N_l} \sigma_y^2 N_l 2^{N_l - 1} \\
=& \sigma_y^2 \frac{N_l}{2} = (\sigma_w^2 y + \sigma_b^2)\frac{N_l}{2}.
\end{aligned}
$$

Iterative application of this result leads to Equation (8), since we have

$$\mathbb{E}\left(|\mathbf{x}^{(L)}|^2 \big| |\mathbf{x}^{(0)}|^2\right) = \int_{\mathbf{y} \in \mathbb{R}_+^L} y_L \Pi_{l=1}^L k_l(y_l, y_{l-1}) \, \mathbf{dy}$$

$$= \int_{\mathbf{y} \in \mathbb{R}_+^{L-1}} \mathbb{E}\left(|\mathbf{x}^{(L)}|^2 \big| |\mathbf{x}^{(L-1)}|^2 = y_{L-1}\right) \Pi_{l=1}^{L-1} k_l(y_l, y_{l-1}) \, \mathbf{dy}$$

$$= \frac{N_L}{2} \int_{\mathbf{y} \in \mathbb{R}_+^{L-1}} (\sigma_{w,L}^2 y_{L-1} + \sigma_{b,L}^2) \Pi_{l=1}^{L-1} k_l(y_l, y_{l-1}) \, \mathbf{dy}$$

$$= \frac{N_L}{2} \sigma_{w,L}^2 \mathbb{E}\left(|\mathbf{x}^{(L-1)}|^2 \big| |\mathbf{x}^{(0)}|^2\right) + \frac{N_L}{2} \sigma_{b,L}^2.$$

where we define $\mathbf{y} \in \mathbb{R}_+^L$ as $\mathbf{y} = (y_1, ..., y_L)$ and $y_0 = |\mathbf{x}^{(0)}|^2$. This leads to Equation (8).

As shown in the proof of Prop. 1, a single component is distributed as $x_i \big| (|\underline{\mathbf{x}}|^2 = y) \sim 0.5\delta_0(\cdot) + \mathbb{1}_{]0,\infty]}(\cdot) p_{\mathcal{N}}\left(\frac{\cdot}{\sigma_y}\right) / \sigma_y$. Thus, its average is

$$\mathbb{E}\left(x_i \big| |\underline{\mathbf{x}}|^2 = \underline{y}\right) = \sigma_y \int_0^\infty p_{\mathcal{N}}(z) z \, dz = \frac{1}{\sqrt{2\pi}} \sigma_y$$

and is identical to Equation (9) according to the definition of $\sigma_y$. The variance reads as

$$\mathbb{V}\left(x_i \big| |\underline{\mathbf{x}}|^2 = \underline{y}\right) = \mathbb{E}\left(x_i^2 \big| |\underline{\mathbf{x}}|^2 = \underline{y}\right) - \mathbb{E}\left(x_i \big| |\underline{\mathbf{x}}|^2 = \underline{y}\right)^2$$

$$= \sigma_y \int_0^\infty p_{\mathcal{N}}(z) z \, dz = \frac{\sigma_y^2}{2} - \frac{\sigma_y^2}{2\pi} = \frac{\pi - 1}{2\pi} \sigma_y^2.$$

To derive the respective expressions conditional on the input, we combine these results with Equation (8). Yet, we cannot compute the average in closed form. But applying Jensen's inequality (for the concave function $\sqrt{\sigma_w^2 \cdot \sigma_b^2}$ yields the simplification

$$\mathbb{E}\left(x_i^{(L)} \big| \mathbf{x}^{(0)}\right) = \frac{1}{\sqrt{2\pi}} \mathbb{E}\left(\sqrt{\sigma_w^2 |\mathbf{x}^{(L-1)}|^2 + \sigma_b^2} \, \big| \mathbf{x}^{(0)}\right)$$

$$\leq \frac{1}{\sqrt{2\pi}} \sqrt{\sigma_w^2 \mathbb{E}\left(|\mathbf{x}^{(L-1)}|^2 \big| \mathbf{x}^{(0)}\right) + \sigma_b^2}.$$

For the variance, we receive exactly

$$\mathbb{V}\left(x_i \big| \mathbf{x}^{(0)}\right) = \sigma_w^2 \mathbb{E}\left(|\mathbf{x}^{(L-1)}|^2 \big| \mathbf{x}^{(0)}\right) + \sigma_b^2.$$

$\square$

## 1.1 Joint signal propagation for multiple inputs

**Proposition 4.** *The same component of pre-activations of signals $h_1, ..., h_D$ corresponding to different inputs $\mathbf{x}_1^{(0)}, ..., \mathbf{x}_D^{(0)}$, are jointly normally distributed with zero mean and covariance matrix $V$ defined by*

$$v_{ij} = Cov(h_i, h_j) = \sigma_w^2 <\underline{\mathbf{x}}_i, \underline{\mathbf{x}}_j> + \sigma_b^2 \tag{11}$$

*for $i, j = 1, ..., D$ conditional on the signals $\underline{\mathbf{x}}_i$ of the previous layer corresponding to $\mathbf{x}_i^{(0)}$.*

*Proof.* Let's assume two different inputs $\mathbf{x}_i^{(0)}, \mathbf{x}_j^{(0)}$ to the same neural network. We study a the same component $c$ of signal pre-activations (at layer $l$) given the previous layer and denote them by $h_i, h_j$. They are given by $h_i = \sum_{k=1}^{N_{l-1}} w_{ck} \underline{x}_{i,k} + b_c$ and correspondingly $h_j = \sum_{k=1}^{N_{l-1}} w_{ck} \underline{x}_{j,k} + b_c$, are thus normally distributed, and depend on the same mutually independent, normally distributed random

variables $w_{ck}, b_c$. As we know, $\mathbb{E}(h_{i/j}) = 0$ so that their covariance conditional on the previous layer is

$$v_{ij} = \text{Cov}(h_i, h_j) = \mathbb{E}\left(h_i h_j\right) = \sum_{k,n} \underline{x}_{i,k} \underline{x}_{j,n} \mathbb{E}\left(w_{ck} w_{cn}\right)$$

$$+ \sum_k (\underline{x}_{i,k} + \underline{x}_{j,k}) \mathbb{E}\left(w_{ck} b_c\right)$$

$$= \sum_k \underline{x}_{i,k} \underline{x}_{j,k} \sigma_w^2 + \sigma_b^2$$

$$= \sigma_w^2 < \mathbf{\underline{x}}_i, \mathbf{\underline{x}}_j > + \sigma_b^2,$$

because of the mutual independence of weights and biases $w_{ck}, b_c$. Since $h_1, ..., h_D$ are jointly normally distributed, they are completely determined by their mean and covariance matrix. $\qquad \square$

**Theorem 5.** *Assume rectified linear units as activation functions. Let two signals $x_1 = \phi(h_1), x_2 = \phi(h_2)$ of the same neuron correspond to two different inputs $\mathbf{x}_1^{(0)}, \mathbf{x}_2^{(0)}$. The variables $y_1 = x_1^2$, $y_2 = x_2^2$, $y_3 = x_1 x_2$ are jointly distributed as:*

$$\begin{aligned} p(y_1, y_2, y_3) =& g(\rho)\delta_{(0,0,0)}(y_1, y_2, y_3) + f_1(y_1)\delta_{(0,0)}(y_2, y_3) \\ &+ f_2(y_2)\delta_{(0,0)}(y_1, y_3) \\ &+ p_W(y_1, y_2, y_3)\mathbb{1}_{R_+^3}(y_1, y_2, y_3) \end{aligned} \tag{12}$$

*conditional on $|\underline{\mathbf{x}}|_1^2$, $|\underline{\mathbf{x}}|_2^2$, $<\underline{\mathbf{x}}_1, \underline{\mathbf{x}}_2>$ of the previous layer. $g(\rho)$ is defined as $g(\rho) = \frac{1}{\sqrt{2\pi}} \int_0^\infty \Phi\left(\frac{\rho}{\sqrt{1-\rho^2}}u\right) \exp\left(-\frac{1}{2}u^2\right) du$ for $\rho \neq 1$ and $g(1) = 1/2$.*

$$f_1(y_1) = \frac{1}{\sqrt{2\pi v_{22}}} \frac{1}{2\sqrt{y_1}} \Phi\left(-\frac{\rho}{\sqrt{1-\rho^2}} \frac{\sqrt{y_1}}{\sqrt{v_{22}}}\right)$$

$$\times \exp\left(-\frac{1}{2} \frac{y_1}{\sqrt{v_{22}}}\right)$$

*is the density of $|\mathbf{x}_1|^2$ for zero $|\mathbf{x}_2|^2$. $f_2(y_2)$ is defined accordingly. $p_W$ refers to the density $p_W(y_1, y_2, y_3) = \frac{1}{4\sqrt{y_1 y_2}} p_{h_1,h_2}(\sqrt{y_1}, \sqrt{y_2})\delta_{\sqrt{y_1 y_2}}(y_3)$, where $p_{h_1,h_2}$ denotes the density of the pre-activations, i..e. two jointly normally distributed random variables with zero means and covariance matrix $V$, while $\rho$ refers to the correlation of the two components: $\rho = v_{12}/\sqrt{v_{11}v_{22}}$. The variables $|\mathbf{x}_1|^2, |\mathbf{x}_2|^2, < \mathbf{x}_1, \mathbf{x}_2 > \big| |\underline{\mathbf{x}}_1|^2, |\underline{\mathbf{x}}_1|^2, < \underline{\mathbf{x}}_1, \underline{\mathbf{x}}_2 >$ are distributed as 3-dimensional convolutions $p^{***N_l}$ and determine the joint signal distribution of the next layer.*

*Proof.* According to the last theorem, the pre-activation signal components $h_1, h_2$ are jointly normally distributed with 2-dimensional covariance matrix $V$ (and zero means). The shape of the joint distribution of $(y_1, y_2, y_3)$ can be deduced from that knowledge. The variables of interest are $(y_1, y_2, y_3) = (0, 0, 0)$ exactly when $h_1 \leq 0$ and $h_2 \leq 0$. This is the case with probability

$$g(\rho) = \frac{\sqrt{1-\rho^2}}{2\pi} \int_{-\infty}^0 \int_{-\infty}^0$$

$$\times \exp\left(-\frac{1}{2}\left(u_1^2 + u_2^2 - 2\rho u_1 u_2\right)\right) du_1 \, du_2$$

$$= \sqrt{\frac{1-\rho^2}{2\pi}} \int_{-\infty}^0 \Phi\left(-\rho u\right) \exp\left(-\frac{1}{2}u^2(1-\rho^2)\right) du$$

$$= \frac{1}{\sqrt{2\pi}} \int_0^\infty \Phi\left(\frac{\rho}{\sqrt{1-\rho^2}}u\right) \exp\left(-\frac{1}{2}u^2\right) du,$$

where the last equality follows from a change of variable if $\rho \neq 0$. The case that only one of the marginals, say $h_2$, is negative, the variables $y_2 = 0$ and $y_3 = 0$ are both zero and the remaining $y_1$

is distributed with density $f_1(y_1) = \frac{1}{2\sqrt{y_1}} p_{x_1}(\sqrt{y_1})$, where $p_{x_1}$ denotes the density of the signal component. $p_{x_1}$ is given by integrating out all cases when $x_2 = 0$, i.e. $h_2 \leq 0$:

$$
\begin{aligned}
p_{x_1}(x_1) &= \frac{1}{2\pi\sqrt{\det(V)}} \int_{-\infty}^{0} \exp\left(-\frac{1}{2\det(V)}(v_{11}x_1^2\right.\\
&\quad + v_{22}h_2^2 - 2v_{12}x_1 h_2)\Big)\, dh_2 \\
&= \frac{1}{\sqrt{2\pi v_{22}}}\Phi\left(-\frac{\rho}{\sqrt{1-\rho^2}}\frac{x_1}{\sqrt{v_{22}}}\right)\\
&\quad \times \exp\left(-\frac{1}{2}\frac{x_1^2}{\sqrt{v_{22}}}\right).
\end{aligned}
$$

The last case, when both $h_1 > 0, h_2 > 0$ and thus also $(y_1, y_2, y_3) > 0$ can be tied to the joint normal distributions of the pre-activations, since $y_1 = x_1^2 = h_1^2, y_2 = x_2^2 = h_2^2, y_3 = h_1 h_2$ then. The given density follows directly from variable substitution.

Since $(|\mathbf{x}_1|^2, |\mathbf{x}_2|^2, <\mathbf{x}_1, \mathbf{x}_2>)$ given the previous layer is the sum of $N_l$ independent random variables that are distributed as $(y_1, y_2, y_3)$, their joint distributions coincides with the $N_l$th convolution of $p(y_1, y_2, y_3)$. $\qquad\square$

**Theorem 6.** *Assume rectified linear units as activation functions. Let two signals $x_1 = \phi(h_1), x_2 = \phi(h_2)$ of the same neuron correspond to two different inputs $\mathbf{x}_1^{(0)}, \mathbf{x}_2^{(0)}$. Let the correlation $\rho = v_{12}/\sqrt{v_{11}v_{22}}$ of the pre-activations $h_1, h_2$ be given, where $V$ denotes the 2-dimensional covariance matrix as defined in Theorem 4. The correlation after non-linear activation is then*

$$
Cor(x_1, x_2) = \frac{\sqrt{1-\rho^2} - 1 + 2\pi\rho g(\rho)}{\pi - 1} \tag{13}
$$

*where $g(\rho) = \int_0^\infty \Phi\left(\frac{\rho u}{\sqrt{1-\rho^2}}\right) e^{-\frac{1}{2}u^2}/\sqrt{2\pi}$. The average of the sum of all components $\mathbb{E}(<\mathbf{x}_1, \mathbf{x}_2>)$ conditional on the previous layer is:*

$$
\mathbb{E}(<\mathbf{x}_1, \mathbf{x}_2>) = N_l\sqrt{v_{11}v_{22}}\left(g(\rho)\rho + \frac{\sqrt{1-\rho^2}}{2\pi}\right) \approx N_l\sqrt{v_{11}v_{22}}\frac{1}{4}(\rho + 1). \tag{14}
$$

*Furthermore, conditional on the signals of the previous layer, $<\mathbf{x}_1, \mathbf{x}_2>$ is distributed as $f_{\mathrm{prod}}^{*N_l}(t)$, where*

$$
\begin{aligned}
f_{\mathrm{prod}}(y_3) &= (1 - g(\rho))\,\delta_0(y_3) + \frac{1}{2\pi\sqrt{\det(V)}}\\
&\quad \times \exp\left(\frac{v_{12}y_3}{2\det(V)}\right) K_0\left(\frac{\sqrt{v_{11}v_{22}}}{\det(V)}y_3\right)
\end{aligned}
$$

*and $K_0(w) = \int_0^\infty \cos(w\sinh(t))\, dt$ denotes the modified Bessel function of second kind.*

*Proof.* The correlation of two random variables can be calculated as $Cor(x_1, x_2) = \frac{\mathbb{E}(x_1 x_2) - \mathbb{E}(x_1)\mathbb{E}(x_2)}{\sqrt{\mathbb{V}(x_1)\mathbb{V}(x_2)}}$. For the correlation between the same signal component corresponding to two different inputs conditional on the previous layer, we can use our insights about the marginal distributions in Equation (9) as

$$
\begin{aligned}
Cor(x_1, x_2) &= \frac{\mathbb{E}(x_1 x_2) - \mathbb{E}(x_1)\mathbb{E}(x_2)}{\sqrt{\mathbb{V}(x_1)\mathbb{V}(x_2)}}\\
&= \frac{\mathbb{E}(x_1 x_2)}{\frac{\pi-1}{2\pi}\sqrt{v_{11}v_{22}}} - \frac{1}{\pi - 1}.
\end{aligned}
$$

After lengthy but not very insightful calculations we obtain

$$\mathbb{E}(x_1 x_2) = \frac{1}{2\pi\sqrt{\det(V)}} \int_0^\infty \int_0^\infty \exp\left(-\frac{1}{2\det(V)}\right.$$

$$\left.(v_{11}x_1^2 + v_{22}x_2^2 - 2v_{12}x_1x_2)\right) x_1 x_2 \, dx_1 \, dx_2$$

$$= \frac{\sqrt{v_{11}v_{22}}}{2\pi}\sqrt{1-\rho^2}\left(\rho g(\rho)\frac{2\pi}{\sqrt{1-\rho^2}} + 1\right),$$

which proofs Equation (13). The average $\mathbb{E}(<\mathbf{x}_1, \mathbf{x}_2>)$ conditional on the previous layer is based on this result:

$$\mathbb{E}(<\mathbf{x}_1, \mathbf{x}_2>) = \sum_{i=1}^{N_l} \mathbb{E}(x_{1,i}x_{2,i}) = N_l \mathbb{E}(x_1 x_2)$$

$$= N_l\sqrt{v_{11}v_{22}}\left(g(\rho)\rho + \frac{\sqrt{1-\rho^2}}{2\pi}\right).$$

Fig. 1 justifies the approximation of this term by $N_l\sqrt{v_{11}v_{22}}\frac{1}{4}(\rho+1)$.

Figure 1: $\mathbb{E}(<\mathbf{x}_1, \mathbf{x}_2>)$ with $v_{11} = 1$, $v_{22} = 1$, $N_l = 1$ in dependence on the correlation $\rho$ of pre-activations according to Eq. (14) (black circles) versus the approximation $\frac{1}{4}(\rho+1)$ (red line).

Next, we have to derive the distribution of $y_3 = x_1 x_2$ conditional on the previous layer. Again, $y_3 = 0$ except for the case $h_1 > 0, h_2 > 0$, thus with probability

$$f_{\mathrm{prod}}(0) = 1 - \mathbb{P}(h_1 > 0, h_2 > 0) = 1 - \frac{1}{2\pi\sqrt{\det(V)}}$$

$$\int_0^\infty \int_0^\infty \exp\left(-\frac{1}{2\det(V)}(v_{11}x_1^2 + v_{22}x_2^2 - 2v_{12}x_1x_2)\right)$$

$$dx_1 \, dx_2 = 1 - \frac{\sqrt{1-\rho^2}}{2\pi}\int_0^\infty \int_0^\infty \exp\left(-\frac{1}{2}(u_1^2 + u_2^2\right.$$

$$\left. - 2\rho u_1 u_2)\right) du_1 \, du_2 = 1 - \frac{1}{\sqrt{2\pi}}\int_0^\infty \int_0^\infty$$

$$\Phi\left(\frac{\rho u}{\sqrt{1-\rho^2}}\right)e^{-\frac{1}{2}u^2} \, du \left(\rho g(\rho)\frac{2\pi}{\sqrt{1-\rho^2}} + 1\right)$$

$$= 1 - g(\rho).$$

The case $y_3 = x_1 x_2 > 0$ is governed by the joint distribution of $x_1 = h_1$ and $x_2 = h_2$ when $h_1 > 0, h_2 > 0$. We note that since $x_2 = y_3/x_1$ with $\frac{dx_2}{dy_3} = \frac{1}{x_1}$, we can change the variable $x_2$ to $y_3$

to obtain the joint density of $x_1$ and $y_3$ as

$$p(x_1, y_3) = \frac{1}{2\pi\sqrt{\det(V)}} \exp\left(-\frac{1}{2\det(V)}(v_{11}x_1^2 + v_{22}\frac{y_3^2}{x_1^2}\right.$$
$$\left. - 2v_{12}y_3)\right)\frac{1}{x_1}.$$

The marginal density of $y_3$ is thus for $y_3 > 0$

$$f_{\text{prod}}(y_3) = \frac{1}{2\pi\sqrt{\det(V)}} \int_0^\infty \exp\left(-\frac{1}{2\det(V)}(v_{11}x_1^2\right.$$
$$\left. + v_{22}\frac{y_3^2}{x_1^2} - 2v_{12}y_3)\right)\frac{1}{x_1}\,dx_1 = \frac{1}{2\pi\sqrt{\det(V)}}$$
$$\exp\left(\frac{v_{12}y_3}{\det(V)}\right)\frac{1}{2}\int_0^\infty \exp\left(-v - \frac{v_{11}v_{22}}{\det(V)}\frac{y_3^2}{v^2}\right)\,dv$$
$$= \frac{1}{2\pi\sqrt{\det(V)}} \exp\left(\frac{v_{12}y_3}{2\det(V)}\right) K_0\left(\frac{\sqrt{v_{11}v_{22}}}{\det(V)}y_3\right).$$

$\square$

## 2 Additional experiments

Figure 2: Time for training $10^4$ steps on MNIST for different initialization schemes, width $N$, depth $L$, with $\sigma_w = \sqrt{2/N}, \sigma_b = 0$. Average time for 100 instances of each configuration is reported along with a $0.95$ confidence interval.

In addition to prediction accuracy, in Fig. 2, we report the time required to train each of the aforementioned networks for $10^4$ steps. Clearly, the width and depth of a network has a high effect on training time. Yet, among networks with the same architecture, our proposed orthogonal $W_0$ initialization scheme allows to train feed-forward neural networks faster, in particular, in case of wider networks with more parameters.

Fig. 3 supports this insight by comparing the learning dynamics of all considered networks on CIFAR-10. Orthogonal $W_0$ initialization achieves a higher accuracy faster (i.e. after a smaller number of training epochs) and more stable, as the variance between different instances is small. While GSM (Gaussian submatrix) initialization outperforms He initialization in these aspects, orthogonal $W_0$ initialization compares still favorable with GSM. Interestingly, shallow layers always outperform

Figure 3: CIFAR-10 test accuracy for different initialization schemes, width $N$, depth $L$, with $\sigma_w = \sqrt{2/N}, \sigma_b = 0$. The mean accuracy and 0.95 confidence interval are reported for 30 trials.

deeper layers for orthogonal $W_0$ initialization at all epochs (on this dataset), while shallow layers train more slowly in the beginning for GSM initialization and then take over deeper layers in performance after $8 - 10$ epochs. However, we have to note that the overall accuracy is far away from the state of the art, as we neither use convolutional layers nor do we apply data augmentation or regularization techniques. In consequence, deeper networks can have the tendency to overfit early. As they also need longer to train in general, we might also not give them long enough time to develop their full potential. We focus solely on the effect of the initialization scheme here and observe that deeper networks become trainable with our initialization scheme than with He initialization.

In summary, the proposed orthogonal $W_0$ initialization allows us to achieve dynamical isometry for deep feed-forward neural networks with ReLU activation and, therefore, improve both accuracy (for deep networks) and training time (for wide networks).

The code for all experiments is provided as additional supplementary material.