[Reviews · NeurIPS 2019]

Reviewer 1



***Additional comments after author response:*** I expressed no major concerns in the review which needed addressing in the author response. The response did elaborate on the relationship between the approaches to ReLU initialization considered and the earlier portion of the paper - this should be made clearer in the paper. However, as pointed out by the other reviewers, the structure in the proposed Gaussian submatrix initalization has previously been proposed in Balduzzi et al. [2]. --- Paper overview: This paper considers the problem of neural network initialization. It analyzes how signals are transformed through the layers of a feedforward neural network, assuming weights are initialized from Gaussian distributions. Previous work used a mean-field assumption to study these dynamics, and used the results to identify parameters for the Gaussians to ensure stable propagation of the mean of the signal variance through the layers, a necessary condition for training deep networks. This work considers how the distribution of the initial signal variance is transformed through the layers of the network. This is done by introducing an integral operator with an activation-dependent kernel to model the transformation. This operator can be viewed as a generalization of the input-output Jacobian considered in other work, and so its spectrum is of interest for understanding stable signal propagation. Having established these results, the authors consider ReLU networks specifically for the remainder of the paper. They derive the integral operator kernel for ReLU, and analyze its spectrum. The analysis shows that there is no way to guarantee stable signal propagation with regular Gaussian initialization in general (i.e. not just in expectation). Further study of the correlation dynamics illustrates that the correlations increase as one moves through the network - as was previously known - but also that narrower networks exhibit higher variance in the correlation distribution. Thus it is possible that networks with narrower layers might be able to more easily capture and maintain small correlations, yielding better trainability. (On the other hand, the authors also point out that broader layers protect against operator eigenvalues larger than one which are problematic for stable gradient propagation.) The authors next propose a novel initialization scheme for deep ReLU feedforward networks. The approach uses parameter sharing to essentially set up the initialized network to behave like a suitably initialized linear network with half as many neurons in each layer. This allows negative correlations to be implicitly captured and propagated using the “duplicated” neurons. The required parameters can now be initialized either through suitable Gaussian initialization or through orthogonal initialization. Empirical results on MNIST and (clipped) CIFAR-10 give preliminary indications the proposed initializations train better at moderate depth than the standard He initialization. Originality: The paper is the first I have encountered that explicitly models the dynamics of the finite-width signal variance distribution. The resulting integral operator and kernel that are studied in the context of ReLU are thus in some sense new objects being studied for the first time. The proposed new ReLU initializations combines a neat parameter-sharing idea with existing approaches to initializing neural networks. Quality: This is high quality work, and I did not detect any noteworthy technical issues. One minor criticism is that the ReLU initialization proposed does not really use the machinery from the earlier part of the paper - even if it might have been inspired by it(?). Furthermore, due to the weight sharing violating the iid assumption on the weights, the traditional theory for analyzing signal propagation is not strictly applicable. As a result, the paper feels a bit like a mix of two papers: a development of some fascinating theory (which could probably do with more details provided), and an empirical study of a proposed initialization scheme which performs well, but arguably could do with a little more theoretical underpinning. The authors also do not mention the (non-)applicability of existing theory in the weight sharing setting, which I think should be discussed. Clarity: The paper is generally well-written; I list some minor corrections at the end of my review. Significance: I think the idea of analyzing the dynamics of the signal distribution could be quite influential, and is likely to be developed further. The proposed initialization scheme certainly seems to warrant larger-scale investigation; if it holds up it may become a new standard for initializing ReLU networks. Other corrections/suggestions: Thm 1: “as transformation by” -> “as transformed by”; l.109: k_l is referred to simply as k L.127: correct “give rise of the signal propagation” L.132 and elsewhere: use \langle and \rangle for inner product L.139: x_1 and x_1 -> x_1 and x_2 Theorem 4: define \delta_0 and p_{\chi_k^2}. L.178 and below: x^m in text vs. y^m in Theorem 4 is somewhat confusing; similarly \lambda_{l,m} vs \lambda_{m,l} - tidy up notation consistency. L.190: it is unclear to me why the eigenfunction converges to the given \delta_0, which seems different to f_{-1} as specified in Thm 4. L.198: this is the inner product, not cosine similarity - as you point it, it is unnormalized Figure 2 caption: diamonds, not squares, would be a better description L.241: Note that the He initialization does not yield stable signal propagation in conjunction with dropout. An alternative initialization scheme [1] indicates one should take the dropout rate into consideration. How does that interact with this conjecture? L.259 : why include \sigma_{w,l} here? l. 271: “reduce a number” -> “reduce the number” Please specify the batch size used for SGD in your experiments? L.278: “He Initialization” -> “He initialization” Ll.291-292: “more parallel” does not make sense. Perhaps: Closer to parallel, more aligned or more correlated. Right hand plot in Figure 4 does not add much, unlike Figure 3 - it may be more valuable using this space for some more details in the text. L.287: “but is not” -> “but it is not” Please improve the references section (Correct capitalization, missing sources such as for reference [4], cite published versions instead of preprints such as for reference [8]). [1] A. Pretorius, E. van Biljon, S. Kroon, H. Kamper. Critical initialisation for deep signal propagation in noisy rectifier neural networks. NeurIPS 2018. [2] D. Balduzzi, M. Frean, L. Leary, J.P. Lewis, K.W. Ma, B. McWilliams. The Shattered Gradients Problem: If resnets are the answer, then what is the question? ICML 2017

Reviewer 2



The authors analyze propagation of low-order moments in artificial neural network, with a particular focus on networks with ReLU activation functions. The difference between the proposed approach and those used in previous works is that they consider a non-asymptotic limit; i.e. the authors do not require the width of each layer to go to infinity. Instead, the authors show that for Gaussian weights ensembles, the distribution of the pre-activations conditioned on the preceding layer post-activation is Gaussian. They then recursively compute the distribution of 2-norms of outputs and expected covariances both conditioned on inputs. The authors analyze the change of measure between two layers using an integral operator and study its spectrum. They use these results to propose an initialization scheme for ReLU networks which allows them to train very deep networks without skip connections and batch normalization. The results in this manuscript are interesting, however their relation to preceding work, both quoted un-quoted ought to be better explained. Firstly, regarding the novelty of the main contribution --- previous work by Pennington et al. has devised isometric and nearly isometric initializations for ReLU networks by shifting the bias appropriately. Another approach, identical to the one taken in this work has emerged from the Shatterew gradients paper [Balduzzi]. While the considerations were not explicitly motivated by mean-field theory, they do consider two point correlations just like this current work, casting doubt about the novelty of this work. Secondly, the relation between the integral operator and the Jacobian matrix used in the previous papers using mean field theory is not made explicit. Both the authors of the work under review and the authors of the mean field papers used a weak-derivative operator but either focused on the change of measure or the treated them as random matrices. It should therefore be stressed that the approach proposed in this work is another interpretation of the same method applied to the same problem. This in no way detracts from the value of the paper, and would only strengthen its connections to existing literature. Finally, a few small issues make it harder to understand the authors points: * Theorem 1 is presented in a super dense fashion * independence and distribution claims are sometimes ambiguous (conditional independence and distribution over random weight distributions?) * N-fold convolution pNl1ϕ(hy) and L-fold application of $T_l$ should be clearly explained * The generalized inverse is not defined $\phi(\cdot)^{-1}$ * Figure two is not clearly labeled. I would suggest changing the colors and/or labeling * Figure 3 $\&$ 4 would benefit from having the legend outside, to make it clearer that the labels apply to both.

Reviewer 3



Originality: I think the proposed initialization is very interesting and seems to work well for fully-connected network with Relu. Could it work well for Conv-nets and perform comparable to ResNet (on cifar10 and imageNet)? Also, several theorems in this paper seems to be known in previous works and it is better to phrase them as lemmas: Thm 2 is standard and has been widely used in recent mean field/overparameterization papers; Thm 3 seems to be similar to Cor 1 page 8 of [HR]; Equations 9 and 10 of Thm 4 seem to be known in [CS]. Quality: the biggest question I have is: can the framework of this paper (theorem 1 and 4) tell us new insight we cannot obtain from recent mean field papers and [HR], [H] and etc.? Also Section 3 and 4 seem to be loosely related to the theory part, i.e. Section 2. They are mostly about dynamical isometry. Could you explain a tighter connection? Clarity: the paper is well-written. Significance: the impact of the paper could be improved if the authors could: 1. using theorem 1 and 4 to obtain new insights about finite width networks; 2. illustrate the success of the initialization method on cifar10 and imagenet. [HR] Boris Hanin and David Rolnick: How to Start Training: The Effect of Initialization and Architecture [H] Boris Hanin Which Neural Net Architectures Give Rise to Exploding and Vanishing Gradients? [CS]:Youngmin Cho and Lawrence K Saul. Kernel methods for deep learning. In Advances in neural information processing systems, 2009.

[Author Response · NeurIPS 2019]

We thank all reviewers for the constructive critique and would be happy to follow their suggestions, in particular,
improve the presentation of our theoretical analysis.

**Insights beyond mean field analysis.** Our analysis provides full distribution information on the joint outputs. All
other works study averages (or usually approx. thereof) and mostly focus on parts of the output distribution only (e.g.,
they either study single inputs or the correlation for two typical inputs). They never provide the full picture and cannot
exclude other parameter choices to improve on the initialization with Gaussian weights. In addition, we explain why the
squared signal norm is the relevant variable to study - because it is the only information that is transmitted from one
layer to the other. While the He initialization preserves this quantity layer-wise *on average*, the distribution becomes
more skewed for increasing depths and the center of max. probability decreases (see Fig. 1 (a)). Furthermore, the
distribution of the cosine similarity explains why moderately deep and wide ReLU networks can be trained despite
negative results by mean field (MF) analysis based on correlations. In addition, we hypothesize that reducing the
effective number of nodes in a layer contributes to the success of DropOut and DropConnect. Next, we explain why an
initialization with parameter sharing leads to improved signal propagation.

A detailed comment for Reviewer #3: Thm. 2 is not difficult to derive but certainly not standard in MF theory. There,
the normal distribution originates from the MF limit. In contrast, here we understand that the output distribution is
completely determined by the empirical covariance matrix of inputs. Higher order moments in the input distribution
have no influence on the output distribution.

**Relation between the theoretical analysis and our initialization proposal.** Our theoretical analysis holds for general
activation function $\phi$. Our specialization to ReLUs identifies several problems that we first try to mitigate by different
parameter choices $\sigma_{w,l}$, $\sigma_{b,l}$. This turns out to be impossible. Hence, we solve it with the GSM by effectively setting
$\phi(x) = x$ at initialization for half of the nodes, while we disregard the other zero half. Note that the entries in $W_0$ are
iid Gaussian and fulfill our assumptions. We could repeat our analysis for linear $\phi$ and show that, e.g., input correlations
are preserved. This is rather obvious however. Instead, we refer to the rich literature on linear neural networks at
initialization. Especially, [SX] suggests the choice of orthogonal $W_0$ for dynamical isometry. We therefore test this
choice in addition.

**Additional literature discussion.** We agree that we have to extend our literature discussion. Closest to our work
is [B] about gradient shattering. Yet, its main focus is on resNets and convNets. We compare with the few results
on fully-connected feed forward layers here. They observe a phenomenon that relates to cosine similarity by an
argument that links signal forward propagation with gradient descent. However, they study it in a setting where they
cannot distinguish between the effect of vanishing gradients and increasing correlations (see Fig. 4 in [B] belonging
to $\sigma^2 = 1/N$). As a result, they observe decreasing correlations, while we have a problem with increasing ones.
Furthermore, they make a claim about the exponentially decaying covariance (Thm. 1 in [B], now with He $\sigma^2 = 2/N$)
without regarding layer width and for typical inputs, while we consider finite layer width and all possible inputs.
"Typical" is defined as required for their proving strategy and seems to be common in networks with batch normalization,
but not in networks without, which we study. Yet, [B] provides a parameter sharing solution similar to ours, but for
convolutional neural networks. It is based on an idea by [S] that was inspired by empirical observations of trained
convolutional filters that learn linear networks. Thus, [B] and [S] provide two additional arguments for the proposed
parameter sharing solution: batch normalization cannot avoid the problems related to shattered gradients and linear
models are good starting points because some layers might need only little adjustment during training.

[HR] and [H] are rigorous studies of the average signal or gradient properties in finite width networks for *single* inputs
to avoid vanishing/expoding gradients, while we study the whole joint output distribution, thus also with respect to
different inputs. In consequence, they do not encounter the problems associated with the cosine similarity. Indeed,
Cor. 1 on p. 8 of [HR] is similar to our Eq. (6). Yet, [HR] assumes the same $\sigma_w$ and $\sigma_b$ on all layers and therefore
does not ask for alternative parameter choices as we do. [CS] studies with Eq. (1) a similar integral as required for our
Thm. 5, but considers the MF limit (in the paragraph after Eq. (8)). [Y] discusses an alternative initialization by shifting
the ReLU with a non-zero bias $b_i$ in a MF context with batch normalization. Initially, we thought about following a
similar approach, but the main problem is that the bias must depend on the input batch and is thus similar to batch
normalization and computationally more intensive.

**[SX]** Saxe et al. Exact solutions to the nonlinear dynamics of learning in deep linear neural networks. ICLR'14.
**[B]** Balduzzi et al. The Shattered Gradients Problem: If resnets are the answer, then what is the question. ICML'17.
**[S]** Shang et al. Understanding and Improving Convolutional Neural Networks via Concatenated ReLUs. ICML'16.
**[Y]** Yang et al. Mean field theory of batch normalization. ICLR'19.
**[HR]** Hanin and Rolnick. How to Start Training: The Effect of Initialization and Architecture. NeurIPS'18.
**[H]** Hanin. Which Neural Net Architectures Give Rise to Exploding and Vanishing Gradients? NeurIPS'18.
**[CS]** Cho and Saul. Kernel methods for deep learning. NIPS'09.


[Meta-Review · NeurIPS 2019]

This paper analyzes the propagation of the distribution of activation variance and correlation through fully-connected feedforward networks, extending a line of prior work that has focused on the mean-field (large width) limit, in which the distributions concentrate around their mean. The reviewers agree that the theoretical analysis is novel and of interest to the community. Particularly noteworthy is the explicit form for the transition kernel for the variance. Some reviewers expressed reservations about the thoroughness of the experimental analysis, but as the main contributions of this work are theoretical in nature, I believe the analysis is sufficient. In the final version, the authors should discuss connections to the looks-linear initialization from Balduzzi et al.